# The Neural Correlates of Developmental Prosopagnosia: Twenty-Five Years on

**DOI:** 10.3390/brainsci13101399

**Published:** 2023-09-30

**Authors:** Valerio Manippa, Annalisa Palmisano, Martina Ventura, Davide Rivolta

**Affiliations:** 1Department of Education, Psychology and Communication, University of Bari Aldo Moro, 70122 Bari, Italy; valerio.manippa@uniba.it (V.M.); annalisa.palmisano@tu-dresden.de (A.P.); m.ventura2@westernsydney.edu.au (M.V.); 2Chair of Lifespan Developmental Neuroscience, TUD Dresden University of Technology, 01069 Dresden, Germany; 3The MARCS Institute for Brain, Behaviour, and Development, Western Sydney University, Sydney 2145, Australia

**Keywords:** congenital prosopagnosia, face processing, brain, neuroimaging

## Abstract

Faces play a crucial role in social interactions. Developmental prosopagnosia (DP) refers to the lifelong difficulty in recognizing faces despite the absence of obvious signs of brain lesions. In recent decades, the neural substrate of this condition has been extensively investigated. While early neuroimaging studies did not reveal significant functional and structural abnormalities in the brains of individuals with developmental prosopagnosia (DPs), recent evidence identifies abnormalities at multiple levels within DPs’ face-processing networks. The current work aims to provide an overview of the convergent and contrasting findings by examining twenty-five years of neuroimaging literature on the anatomo-functional correlates of DP. We included 55 original papers, including 63 studies that compared the brain structure (MRI) and activity (fMRI, EEG, MEG) of healthy control participants and DPs. Despite variations in methods, procedures, outcomes, sample selection, and study design, this scoping review suggests that morphological, functional, and electrophysiological features characterize DPs’ brains, primarily within the ventral visual stream. Particularly, the functional and anatomical connectivity between the Fusiform Face Area and the other face-sensitive regions seems strongly impaired. The cognitive and clinical implications as well as the limitations of these findings are discussed in light of the available knowledge and challenges in the context of DP.

## 1. Introduction

Faces represent the stimuli we rely on the most for social interactions. They provide cues on others’ identity, age, gender, attractiveness, race, approachability, and emotions. Based on the evolutionary relevance of faces, most individuals can recognize others’ identities effortlessly thanks to dedicated face-selective cognitive mechanisms and the respective neural substrates [1]. In fact, unlike most everyday items, human faces are perceived as a whole, rather than assortments of features. According to the “domain-specific hypothesis” faces are processed holistically either due to an innate facial template [2] or because human faces represent the sole uniform stimuli for individual-level discrimination during the sensitive developmental period [3]. Alternatively, according to the “expertise hypothesis”, holistic processing results from automatized attention to whole objects, which is developed with extensive experience in discriminating them [4]. Despite the pivotal role of holistic processing, to date, the more consistent hypothesis (i.e., the “featural/configural hypothesis”) postulates that faces are processed by using both holistic (configural) and featural (i.e., analytic) mechanisms, emphasizing global characteristics (i.e., spatial relations) and specific face components (i.e., eyes, nose, and mouth), respectively [5,6,7]. Holistic and featural analyses of face stimuli can be used alternatively or together based on the task requests, stimulus presentation, and contextual demands [8,9].

The impairment of either or both of these mechanisms plays a role in a specific condition known as prosopagnosia [10], characterized by serious and (often) specific face identification deficits [11,12,13]. Albeit early scientific reports refer to the *acquired* form of prosopagnosia, where people lost their previously intact face recognition ability after a brain injury, research over the last ~25 years has increasingly focused on the *developmental* (or congenital) form of prosopagnosia (DP), which refers to the lifelong difficulty in recognizing faces despite the absence of brain damages [7,14,15,16]. 

Along with studies assessing the cognitive phenotype and interindividual variability of developmental prosopagnosia [14], much research has focused on the neural underpinnings of this condition [17]. This corpus of research is based on evidence from neurotypical individuals, showing the existence of face-sensitive neuro-cognitive mechanisms. Functional magnetic resonance imaging (fMRI) and invasive neuronal recordings in humans (and non-human primates) reveal the existence of a network of face-sensitive regions in the ventral occipito-temporal cortex (VOTC) [18,19,20,21,22]. Face-sensitive regions are mostly (albeit not specifically) right-lateralized and identified in the lateral surface of the inferior occipital cortex (i.e., occipital face area—OFA) [23], the lateral side of the fusiform gyrus (i.e., fusiform face area—FFA) [19], and the posterior part of the superior temporal sulcus (pSTS) [24]. 

Further neuroimaging evidence has unveiled two main face-sensitive networks in the human brain: (i) the *core face network* and (ii) the *extended face network* [21,25]. Following the early-stage analysis of face and structural processing in the OFA [26], information on invariant facial features (i.e., crucial information for face identity) reaches the FFA [21,27], while dynamic (changeable) features such as movements in eye gaze or facial expressions are directed to the pSTS [28,29,30]. Additional face-sensitive regions have also been described in the so-called extended face network. Indeed, the FFA projects to the anterior part of the medial (aMTG) and inferior temporal gyrus (aITG), which process the biographical and semantic information of known faces [31,32] (i.e., as suggested by patients with aTC damage causing the inability to access person-specific information from faces and names [33,34,35]). Further engaged areas include the intraparietal sulcus, which directs attention based on gaze; the auditory cortex, involved with speech perception; and the amygdala and limbic system, which process emotional information [20,21].

As for the temporal dynamics of face-processing, event-related potentials (ERPs) show that faces elicit specific occipito-temporal components [36]. A well-established ERP marker of face-sensitive cortical processing is the N170 [37,38,39], which consists of a large and often right-lateralized electroencephalography (EEG) deflection peaking between 150 and 200 ms over the occipitotemporal cortex in response to faces compared to non-face stimuli [40]. In magnetoencephalography (MEG), a similar component (i.e., the M170) is also observed [41,42,43,44]. Other components involved in face processing include the P1 (indexing a very early stage of face processing [45,46]), the N250 (reflecting the activation of preexisting and acquired face representations [47,48]), and the P600f (indexing later post-perceptual stages of face recognition [38,49]).

Albeit early neuroimaging—mainly single case—studies failed to show functional and morphological abnormalities in individuals with developmental prosopagnosia’s brains [50,51], recent evidence has shown face-network abnormalities at multiple levels [52,53]. This scoping review aims to shed light on convergent and contrasting findings, reviewing twenty-five years of neuroimaging literature on the anatomo-functional correlates of developmental prosopagnosia. Our objective is to map the functional and anatomical differences between individuals with developmental prosopagnosia and healthy controls to provide a starting point for future studies in this field.

## 2. Literature Search

Two researchers (AP and VM) conducted a literature search using PubMed (URL: https://pubmed.ncbi.nlm.nih.gov/, accessed on 7 May 2023) and Web of Science (URL: https://www.webofscience.com/, accessed on 7 May 2023) for reports published in the English language without time limits (last updated on 1 May 2023). The search keywords included (“developmental prosopagnosia” OR “congenital prosopagnosia”) AND (“magnetic resonance imaging” OR “diffusion tensor imaging” OR “electroencephalography” OR “magnetoencephalography” OR “positron emission tomography”). During the screening of publications, we also searched within their reference lists to identify additional eligible studies. Only original research comparing the brain structure and activity of healthy control participants (HC) and individuals with developmental prosopagnosia (DPs) were included in this review. In Table 1, the total number of records and studies included in this review are summarized based on technique and DPs sample size. The overall sample of DPs was 818, but many of them were involved in multiple studies. Due to the high heterogeneity in the methods, procedures, outcomes, sample selections, and study designs, we deliberately opted for a *scoping review* [54,55]. The methodologies, sample characteristics, and main results of the included studies are summarized in Table 2 (MRI studies) and Table 3 (EEG and MEG studies).

## 3. The (In)visible Brain Markers of Developmental Prosopagnosia

### 3.1. Gray and White Matter Alterations

Contrary to the acquired form, a DP diagnosis requires face-processing difficulties to be present (presumably) since birth, not caused by any sign of a brain lesion, together with normal sensory and intellectual functions [103]. Nevertheless, differences in the cerebral architecture and connectivity have been reported in DP. Using various MRI techniques such as structural MRI, diffusor tensor imaging (DTI), and functional connectivity fMRI, researchers have been able to investigate the differences between DPs and HCs. This has allowed for an analysis of the links between structural and behavioral data.

Most of the reviewed studies reported a reduced density or volume in DPs’ temporal lobes compared to HCs, specifically in the pSTS, MTG, and FG (e.g., [21,25]). Such evidence was more consistent within the right hemisphere. As for white matter integrity, lower fractional anisotropy and functional connectivity were found in the core face network, particularly near the r-FFA (e.g., [53,75]). These findings are consistent with studies on the neural basis of face processing [104] and with injuries reported in acquired prosopagnosia [105,106]. On the other hand, the MTG and ITG are not face-selective regions; despite this, they are implicated in identifying and naming famous faces and buildings (i.e., semantic memory) [107].

Albeit DPs’ FFA and OFA gray matter volumes do not seem to be reduced, there is converging evidence about disrupted white matter within DPs’ VOTCs. Particularly, the r-OFA and r-FFA have emerged as central nodes where the connectivity within the core face network is compromised in DPs. Specifically, the r-OFA shows impairment in both the short-range and long-range functional connectivity within the core face network, whereas the r-FFA shows impairment mainly in the long-range functional connectivity and within the extended face network [52,53,79]. Further analyses revealed multiple regions in DPs’ core- and extended-face networks, whose functional connectivity to the r-OFA and r-FFA is decreased; such findings suggest the central role of the interaction between the r-FFA and the other regions of the core face network and the extended face network to successfully recognize faces. 

To summarize, despite the absence of a brain injury, gray matter alterations in the temporal lobe and white matter reductions involving the VOTC and aTC seem to characterize the DPs’ brains (see Figure 1). Reduced r-FFA’s white and gray matter volume and short-range functional connectivity would explain DPs’ face perception deficits, whereas disrupted functional connectivity between the r-FFA and r-pSTS and r-aTC seems to explain DPs’ deficits in face learning (i.e., memory) [53,73]. This hypothesis is in line with Garrido et al. [82] that the grey matter volume in the l-pSTS, aMTG, and r-FG (reduced in DPs compared with HCs) correlates with facial identity scores. Correlations between FFA structural deficits and behavioral measurements are observed mainly in the right hemisphere, where a dominance for face processing has been suggested by its larger size [19,108], higher probability of occurrence [109,110], and higher anatomical localization consistency of face-selective regions [111] compared to the left hemisphere. 

Some results, however, do not support this conclusion. For instance, Haeger et al. and Gilaie-Dotan et al. [61,77] did not find any structural differences in DPs, while Behrmann et al. found larger MTGs in DPs [80]. Thomas et al. [52] reported a marked reduction in the structural integrity of two long white matter tracts in DPs, but subsequent studies have not confirmed or refined these results. Finally, a link between face memory impairments and decreased cortical density in the left lingual gyrus and l-DLPFC was found by Dinkelackler et al. [58]. Despite the role of these two regions in both visual cognition and memory processes, we point out that Dinkelackler’s DPs also showed mild impairments in non-face visual memory. Therefore, this finding cannot be attributed solely to deficits in facial memory but may reflect a more general visual impairment.

### 3.2. Face-Induced Brain Activity in Developmental Prosopagnosia

The studies investigating face-induced neural activity in DPs compared to HCs show different methodologies and findings (see fMRI studies in Table 2). The low temporal resolution of blood-oxygenation-level-dependent (BOLD) signals from fMRI, the advantage in terms on signal-to-noise ratio, and the technical limitations of performing tasks in the scanner determined a bias towards block designs compared to event-related designs. Indeed, most of the fMRI evidence for DP comes from block designs, particularly those adopting passive viewing and one-back tasks of faces vs. non-face stimuli. Although the former task is a perceptive task and the latter is a sequential face-discrimination task involving low memory load, they were both used to investigate the neural bases of face processing in DPs. 

First, Jiahui et al. [63] investigated how selective attention to different aspects of faces affects brain activity in HCs and DPs; their results show that attention towards specific facial features (i.e., selective attention) modulates activity in both ventral areas (OFA and FFA) and dorsal areas (pSTS and inferior frontal gyrus): the modulation profiles in both pathways are similar between HCs and DPs, suggesting that DPs’ difficulties with recognizing faces are not due to attentional alterations, but rather due to face-specific perceptive or memory deficits. Accordingly, FFA activity when viewing faces correlates with individual differences in baseline face identification performances [59], and a lower FFA and OFA activity, mainly in the right hemisphere, was indeed observed in DPs during face viewing [66,67,69]. Moreover, DPs exhibited increased activity in the posterior visual regions and decreased connectivity between the occipital areas and the anterior temporal and frontal regions when viewing faces in comparison to HCs. These alterations correlate with face recognition scores and suggest the presence of peculiar network patterns in DPs [68]. 

To date, the model that best explains how face-relevant information flows through face-selective areas is based on the presence/absence of faces, which modulates the feed-forward effective connectivity from the primary and secondary visual cortices to the core face network. The connectivity within these networks during face viewing is significantly diminished in DPs relative to HCs, indicating that these connections may contribute to typical face-selective responses as well as accurate facial recognition [64]. Overall, these studies provide valuable insights into the neural mechanisms underlying face processing in DPs and highlight the prominent role of the FFA and its connections with other brain regions in face recognition.

Some researchers attempted to dissociate the neural activity related to face memory and perceptual processing; although DPs experience deficits in both face perception and face memory, there is a weak correlation between their performances in these tasks, indicating dissociable neural correlates. Compared to HCs, DPs show separate neural correlates for face memory and face perception within the core face network. Particularly, face memory is associated with activation in the bilateral FFA, while face perception is linked to face selectivity in the r-pSTS [72]. For instance, by adopting a modified Sternberg paradigm, Haeger et al. [77] investigated FFA activation patterns during face memory encoding and maintenance. An increased memory load entailed higher FFA activation and a higher degree of correlation between the activated voxels in HCs but not in DPs. Furthermore, the FFA activation patterns in the DPs were more unstable across the trials compared to the HCs. These findings suggest that DPs exhibit altered brain responses during the encoding and maintenance of face stimuli, which is linked to reduced performance in both long-term memory and mental imagery tasks with faces. Congruently, when learning of unfamiliar faces, DPs exhibit neural deficits characterized by diminished repetition suppression for faces in the FFA and decreased pattern stability for repeated faces in the bilateral MTG [71]. This indicates impairments in the perceptual analysis in the FFA and disrupted propagation from the FFA, which, along with deficits originating from the MTG itself, result in unstable mnemonic representations in the MTG. Notably, a significant correlation between memory task performance and pattern stability is found in the left MTG, but not the right MTG. 

Other studies reported significant alterations beyond the VOTC. For instance, abnormalities in neural activity in response to familiar compared to unfamiliar faces in DPs exist in the left precuneus, anterior, and posterior cingulate cortex [76]. Furthermore, Rivolta et al. reported reduced face sensitivity in the aTC and reduced face–object discrimination in the right parahippocampal gyrus [69]. These regions are part of the extended face network and linked to post-perceptual face-processing stages, such as encoding or the retrieval of semantic and episodic memories about specific individuals [31]. Another study reported a reduced fMRI signal during face passive viewing in DPs’ LOCs, a region involved mainly in object visual processing [66], while Avidan et al. [51,56] failed to find any differences in brain activity between DPs and HCs during the passive viewing of faces and non-face stimuli.

To sum up, the reviewed evidence suggests that DP is characterized by abnormal face representations that differ qualitatively from HCs. Indeed, DPs may rely on different aspects of facial features for successful face recognition (i.e., parts-based strategies) due to the FFA’s grey and white matter disruptions. Moreover, DPs who exhibit typical face perception performance show differential activation patterns compared to HCs, suggesting that some DPs develop compensation strategies. Overall, DPs exhibit abnormalities in neural face representation in both the FFA and r-OFA, indicating difficulties in both holistic and featural face processing (see Figure 2). The presence of both shared and specific neural correlates of face memory and face perception may help explain the heterogeneous nature of DPs and the related findings. However, several questions need to be addressed. First, while face perception is traditionally considered a preceding stage of face memory, the weak correlation between behavioral performances and the distinct core face network neural correlates of face perception and memory suggest that the former might be relatively independent of face memory. Secondly, deficit comorbidity in face memory and face perception in the extended face network is consistent with the hypothesis that the extended face network integrates information from the core face network. Future fMRI studies, using a simultaneous matching paradigm, should dissociate face perception and memory processes, investigating the role of these two cognitive domains in a developmental prosopagnosia deficit.

## 4. EEG and MEG

The evidence from the EEG/MEG in DPs is summarized in Table 3. The functional impairment of the face-processing systems in DPs has been largely investigated via ERPs, which allow for examining real-time brain dynamics underlying face processing [36]. The N170—occurring around approximately 170 ms at the right lateral temporal electrode sites following the presentation of facial stimuli—represents a reliable marker of the early activation of facial representations [38,112], when features are perceptually “glued” into an indecomposable holistic whole [49,113].

Some ERP studies provide evidence for abnormal N170 responses in DPs. Since the first evidence from [81], the non-specificity of this component in DP has been reported in multiple studies showing that DPs exhibit same-amplitude N170 in response to faces and objects [50,61,84,87], as well as no typical N170 right laterality [86,98]. Larger-than-normal noise elicited from the N170 might account for the abovementioned non-selectivity of this component in DPs [90]. Indeed, early encoding processes reflected in the N170 might fail to discriminate between face and non-face stimuli in DPs [81,90]. As face identification relies on configural encoding, an impairment of this mechanism might be compensated by other general encoding strategies, resulting in poor face recognition and analogous electrophysiological responses to faces and non-face stimuli [84]. 

Evidence for reliable N170 amplitude differences between faces and nonface stimuli in DPs has also been reported [89]. It should be noted that neurophysiological discrepancies among studies might stem from differences in diagnostic criteria and participants’ characteristics (e.g., heterogeneous performances among DPs). This supports the idea that DP reflects a heterogonous impairment (i.e., with face-processing deficits being on a continuum) and that some behavioral deficits might not necessarily correlate with a lack of the face-selective N170 [114]. In addition to inter-individual differences, a compensatory mechanisms might be adopted by DPs to face perceptual processing deficits [89]. This implies the need for adopting multiple (e.g., both perceptual and mnemonic) and consistent measures of face processing across studies.

While face inversion in HCs leads to an enhanced and delayed N170 due to a loss of configural information [115,116], no such effect of upside-down faces is reported in DPs [91]. This could represent a functional deficit in configural face processing (i.e., less efficient use of prototypical spatial–configural information provided by upright faces). Indeed, DPs might prominently rely on feature-based strategies when processing faces [92]. In line with the idea of a reduced face-selectivity of visual areas in DPs, the paradoxical larger N170 for upright stimuli reported in some DPs might stem from the activation of object-sensitive areas in response to faces [91]. Interestingly, this abnormal inversion effect is reported with body stimuli, potentially due to the commonalities between face and body perception [65] and the almost equal relevance of configural mechanisms for body and face processing [117,118]. 

ERP evidence in DPs includes components such as the P1 (indexing a very early stage of face processing) [45,46], the N250 (reflecting the activation of preexisting and acquired face representations) [47,48], and the P600f (indexing later post-perceptual stages of face recognition) [38,49]. While evidence from the P1 component shows no inversion effect in DPs [92], N250 responses are reported for non-recognized faces in some DPs [95]. However, they seem to be attenuated, delayed, or qualitatively different in DPs compared to HCs [93,94,96]. These results suggest that stored visual representations of known faces might be available for DPs [119]. However, the activation of identity-specific visual memory traces could be inefficient in DPs and lead to covert face recognition. The absence of a P600f component for non-recognized famous faces coupled with reliable N250 indicates that some DPs might exhibit face recognition impairments at a later stage of face processing, potentially due to the disruption of links between stored visual representations of faces and semantic or episodic representations in long-term memory [95,120,121]. Furthermore, no P3 responses (i.e., reflecting advanced stages of face processing) are reported in DPs [98], suggesting that information for face representations could not be sufficiently attended to or deeply encoded. Despite the atypical patterns of visual scanning that cannot be excluded, these results provide support for the engagement of insufficient encoding resources and idiosyncratic cognitive strategies for face processing.

Further support for the abnormal electrophysiological responses in DP include the attenuated neural responses to unfamiliar faces in DPs compared to HCs (assessed via Fast Periodic Visual Stimulation EEG) [97], as well as abnormal and delayed EEG responses to faces (i.e., similar to those for processing non-face stimuli) [85]. This highlights the engagement of different cognitive processes during face recognition between DPs and HCs, with the former relying on a pathway more commonly associated with objects [85].

As compared to EEG, MEG provides good spatial resolution to investigate the neural sources of face-elicited responses [122]. Multiple studies have found a strong magnetic response (M170) to face stimuli compared with non-face stimuli over occipitotemporal brain regions in HCs [123]. The neural generators of the M170 are identified within the VOTC (e.g., [124]). Face-selective M170 patterns are reported in DPs, suggesting that impaired face recognition in developmental prosopagnosia is not necessarily characterized by an absence of face-specific responses [102]. Furthermore, in Rivolta et al. [100], DPs’ face-selective M170 in r-LOC correlates with configural face processing, whereas rFG-M170 correlates with featural processing. Importantly, M170 sensitivity correlates positively with face detection performances. 

In addition to M170 abnormalities, some studies report alterations in brain activity in the ‘gamma’ frequency band (i.e., >30 Hz) in DPs [88,101,125]. This is in line with the literature highlighting the role of gamma-band oscillations at multiple stages of face perception [126,127,128] and the evidence for induced gamma-band responses as electrophysiological markers of face processing [99,129]. Again, the electrophysiological patterns of brain responses to faces in DPs, as well as the adopted cognitive strategies, might be qualitatively different across DPs and account for the literature inconsistencies. Indeed, genetically based prosopagnosia does not refer to a single trait, as it can encompass a cluster of subtypes (i.e., with patterns of impairments in specific components of the face-processing system) in individuals from the same family [130].

Despite DPs’ face perceptions being associated with both typical and atypical brain responses, the activity in face-sensitive areas seems to be altered in DPs compared to HCs, with (i) electrophysiological markers demonstrating the occurrence of overt face processing, (ii) differential activity patterns (e.g., laterality), and (iii) the activation of object-sensitive areas in response to faces, suggesting the adoption of insufficient or inadequate strategies. The main reliance on feature-based processing mechanisms and the lack of configural strategies seem to be consistent across studies. The literature inconsistencies might be related to differential participants’ selection procedures and adopted methods, which implies the need for defining more consistent diagnostic criteria for DP.

## 5. Literature Weakness

As reported in this review, different studies present sparse or contrasting findings. This divergence can be explained by several factors. First, the sample size in more than 35% of the reviewed literature accounts for five or fewer DPs (see Table 1), with most of them being single cases. Most DPs exhibit heterogeneity in their symptom manifestation and have been involved in multiple assessments using different techniques and procedures (e.g., [67,71] or [80,103]). 

The selections of DPs represents a major issue in this field due to the lack of consensus on the exact diagnostic procedures and criteria [131]. Currently, researchers primarily rely on a limited set of neuropsychological tests investigating face processing. The most widely adopted are the Cambridge Face Memory Test [14], the famous faces test, and the Cambridge Face Perception Test [132]. However, arbitrary criteria are adopted to recruit DPs, and they vary among studies (e.g., in the adopted tests and versions and cut-off scores). In some studies, a diagnosis of DP is also provided to individuals exhibiting agnosia for objects other than faces, and comprehensive neuropsychological (e.g., including intelligence quotient) and vision disorder assessments are, in most cases, not provided [133]. We point out that standardizing the diagnostic criteria is crucial for the proper interpretation of the neuroimaging and neurophysiological findings.

Together with the intrinsic heterogeneity of DP, variations in the sample compositions and methods might have led to potential biases and could explain at least some of the inconsistencies. These methodological and procedural issues have been addressed in the recent literature with the adoption of larger samples and more rigorous protocols. For instance, most of the fMRI studies reviewed in this article used a block design instead of an event-related design, since the former provides a higher signal-to-noise ratio. Further, in recent years, fMRI protocols have embraced multivariate analyses (e.g., MVPA), which provide a more sensitive analytical approach than a traditional univariate analysis [134,135]. These studies have corroborated the involvement of the aTC in face processing and DP [69,70,71], a region known to be susceptible to fMRI signal distortion and drop-out [136,137].

## 6. Conclusions

DP is a neurodevelopmental disorder in which brain structural, functional, and electrophysiological alterations are observed. Consistent with the strong heritability of face recognition in the general population, DP has a genetic component (precisely, it may be a monogenic, autosomal dominant disorder) [132,138,139]. However, little is known about its onset, which is thought to be heterogeneous (i.e., of multiple etiologies). Similarly to other selective neurodevelopmental conditions, one hypothesis involves neural migration errors in the occipital and temporal regions during brain development [140,141]. This would also account for DP’s heterogeneity, based on how circumscribed these errors occur. However, given the lack of evidence, DP etiology represents a matter of debate [16]. 

The available MRI evidence highlights some recurring patterns in DP. Reduced gray matter volume is often observed in DPs’ temporal lobes, specifically in the pSTS, MTG, and FG. White matter alterations have been found in the core face network, particularly near the r-FFA. fMRI studies assessing brain activation in response to faces indicate that DPs exhibit lower activity in the right FFA and right OF, potentially due to disrupted feed-forward connectivity from the visual cortices to the core face network. The predominant right-lateralization of the impairments is in line with the evidence supporting the right hemisphere’s dominance in face processing [142]. Neural activation abnormalities in DP also extend beyond the VOTC to regions involved in post-perceptual face processing and object visual processing. Evidence from EEG/MEG studies reveal atypical N170 responses, with non-specificity and a lack of right laterality. Given the methodological and sample selection differences among the reviewed studies, the available findings are not fully consistent. Inter-individual variability among DP patients might also account for these inconsistencies and be linked to the DP heterogeneity.

Although most of the structural and functional impairments observed in DPs primarily involve the right hemisphere, the involvement of the left hemisphere regions is also common. Indeed, the neural face-processing network is distributed across both hemispheres, although a relative right-hemispheric dominance has been predominantly reported [143]. For example, Thome et al. [144] used fMRI to evaluate the cerebral face perception network in 108 healthy adults. While the average brain activity was higher in right-hemispheric areas than in left-hemispheric regions, this asymmetry was rather mild when compared to other lateralized brain functions such as language and spatial attention. This asymmetry differed greatly across individuals. The differences in lateralization between the core face network regions were not significant, and left-handed people did not display a general leftward shift in lateralization. However, when compared to right-handed men, left-handed men demonstrated a pronounced left-lateralization in the FFA. Another recent study [145], employing lateralized Rubin’s vase–faces figures, revealed that when the figure is presented in the left visual field (and processed by the right hemisphere), face perception (over vase) is more prevalent in males than in females. This difference is likely attributed to the stronger (right) hemispheric dominance observed in males compared with females when decoding face stimuli [146]. These findings highlight the heterogeneity in the individual patterns of hemisphere dominance for face perception and the importance of investigating the role of both hemispheres in DP.

These structural and functional alterations lead to face recognition and learning difficulties in DPs. To date, three main hypotheses have been proposed to explain the face-processing impairment in DPs: (i) the inefficient use of cognitive mechanisms devoted to face processing [147]; (ii) the impairment of within-class discrimination mechanisms that are not specific to faces [148,149]; and (iii) the reliance on different neurocognitive mechanisms to HCs, (i.e., with faces processed similarly to non-face stimuli) [150,151,152]. This latter hypothesis, which does not exclude an inefficient use of face-specific cognitive mechanisms, is extensively supported by the cognitive literature on DPs’ reliances on atypical aspects of facial features for successful face recognition [67]. This could reflect an adaptive mechanism in response to morphological and functional brain alterations.

The behavioral deficits in DPs have been strictly linked to the FFA white and grey matter abnormalities, as demonstrated by reduced face-selective activity in both the OFA and FFA. Indeed, although OFA gray matter seems not to be affected, the disrupted functional connectivity between the OFA and FFA could contribute to the normal reconstruction of individual facial features in a holistically integrated configuration, resulting in more feature-based processing [70,152,153]. Furthermore, the disrupted functional connectivity between the FFA and pSTS might contribute to the compromised static and dynamic integration of facial features.

At the behavioral level, not all DPs display reduced accuracy in face perception tasks, while face learning and memory are always impaired [154,155,156]. The use of inefficient face-processing strategies might interfere with the encoding of face identity as well as semantic and biographical information. EEG and MEG data provide support for this hypothesis.

Some relevant considerations can be drawn for future directions in this field. Future studies should select homogeneous samples of DPs based on an accurate assessment of their behavioral manifestations to account for disease heterogeneity. Different aspects of face processing (i.e., recognition, memory, discrimination) and face features (identity, expression, gaze) should be assessed simultaneously to uncover systematic associations and dissociations between different face deficits, which will unveil the varied behavioral profiles of face recognition deficits. In terms of interventional trajectories, the holistic face training developed by DeGutis et al. [157,158] was found to be effective in improving face identification, N170 face-selectivity, and functional connectivity between the r-FFA and r-OFA in DPs. These findings suggest that certain training regimes may improve face recognition ability in DPs. However, future studies should investigate therapeutic options in randomized controlled trials and assess the generalizability of training to daily life [16]. Indeed, the current literature lacks evidence on effective interventions to ameliorate DPs’ deficits. It would also be beneficial to develop a valid taxonomy of DP, which will help resolve inconsistent findings and facilitate research.

## Figures and Tables

**Figure 1 brainsci-13-01399-f001:**
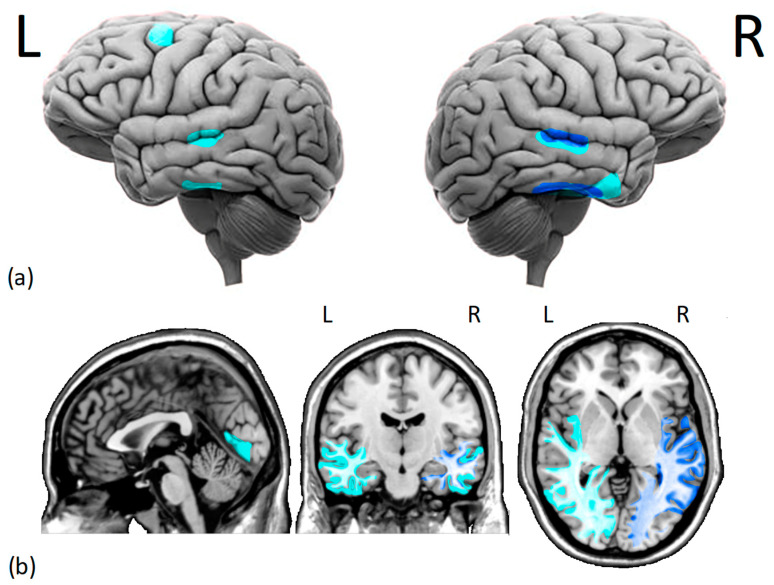
Cortical (**a**) and subcortical (**b**) issues in the gray and white matter of DPs brains. Several studies have found (i.e., dark blue) reduced morphometry measurements in the right fusiform face area and right superior temporal sulcus of DPs. Additionally, there is moderate evidence (i.e., light blue) of decreased gray matter density in the middle temporal gyrus, right anterior inferior temporal gyrus, left dorsolateral prefrontal cortex, and lingual gyrus. Moreover, there is a reduction in white matter integrity and functional connectivity within the core face network and middle-anterior temporal cortex, especially on the right hemisphere. L = left hemisphere, R = right hemisphere.

**Figure 2 brainsci-13-01399-f002:**
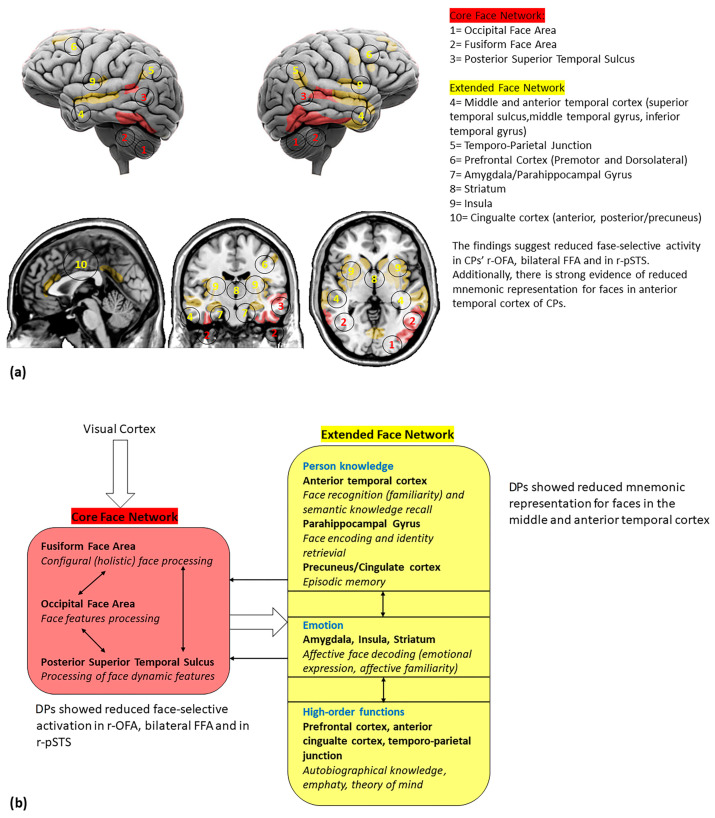
(**a**) Cortical and subcortical brain regions involved in face perception; (**b**) a reviewed model of face networks, including the main findings from DPs studies. The white bold arrows indicate the links between different networks and the thin black arrows indicate the links between network hubs (adapted by Haxby et al. [21]).

**Table 1 brainsci-13-01399-t001:** Outline of DP’s sample sizes per technique in the reviewed literature. The number of papers included in the review was 55, with 6 of them including 2 techniques for a total of 63 studies. Abbreviation: sMRI = structural MRI; DTI = diffusion tensor imaging; fc-fMRI = functional connectivity fMRI.

DPs Sample Size	sMRI N	fMRI N	DTI N	fc-fMRI N	EEG N	MEG N	Tot N	%
Single case	4	4	1	0	5	0	14	22.22
≤5	0	4	0	0	4	1	9	14.29
6 ≥ 10	1	5	1	1	4	3	15	23.81
11 ≥ 20	2	5	2	1	6	1	17	26.98
≥21	1	4	0	3	0	0	8	12.70
Tot	8	22	4	5	19	5	63	100.00

**Table 2 brainsci-13-01399-t002:** Magnetic resonance image studies comparing brain morphology, connectivity, and activity in developmental prosopagnosia participants (DPs) and healthy controls (HCs). Task abbreviations: FFT = Famous Face Test; CFMT = Cambridge Face Memory Test; CFPT = Cambridge Face Perception Test; BFRT = Benton Facial Recognition Test; WFMT = Warrington Face Memory Test; RMT = Recognition Memory Test; IQ = intelligence quotient.

Protocol/ Outcomes	Reference	Participants	Task Administered during the Protocol	Main Findings	Diagnosis and Behavioral Data
Block fMRI	Avidan et al., 2005 [51]	4 DPs, 10 HCs	(1) One-back task (face vs. object); (2) motion pictures passive viewing (face vs. object); (3) adaptation (face vs. object); (4) Rubin’s face–vase	DPs showed normal fMRI activation in the FFA and the rest of VOTC during both face and non-face stimuli viewing. DPs also showed normal adaptation levels like HCs and exhibited evidence of global face representation in the FFA.	DPs scored significantly lower than HCs in recognizing familiar faces and in the same/different discrimination judgments on unfamiliar faces. DPs were also impaired, although to a lesser extent and with greater variability, on tasks involving nonface stimuli. All DPs performed within the normal range on various low-level visual processing tasks.
Block fMRI	Avidan et al., 2014 [56]	7 DPs, 7 HCs	Passive viewing (emotional faces, neutral faces, famous faces, unfamiliar faces, or buildings)	DPs showed normal activation patterns and functional connectivity in the core face networks and the amygdala.	DPs reported substantial life-long difficulties with face processing. DPs scored below (2 SD) the normal range on at least 2 out of 4 different diagnostic tests (FFT, CFMT, CFPT, discrimination of novel upright and inverted faces).
Block fMRI	DeGutis et al., 2007 [57]	1 DPs, 24 HCs	One-back task (faces, scenes); face classification training	The DP case showed face-selective activation in bilateral FFA and r- OFA. After the training, the functional connectivity between FFA and OFA increased.	The DP case was severely impaired in face recognition and classification, whereas she performed within the normal range on one-back face task and in object and word recognition. Following training, DP case’s performance at the face classification task improved.
Block fMRI	Dinkelacker et al., 2011 [58]	24 DPs, 25 HCs	Discrimination task (neutral, positive, negative emotional faces, building, and scrambled faces)	DPs showed decreased signal during face processing in the FG, LOC, and right DLPFC. Processing of buildings was accompanied by decreased brain activity in the MTG, ventral, and precentral DLPFC.	DPs were impaired in long-term memory for faces and, to a lesser extent, other complex visual stimuli such as buildings. However, their memory for faces with negative valence and the ability to categorize emotional expressions were preserved.
Block fMRI	Furl et al., 2011 [59]	15 DPs, 15 HCs	Passive viewing (cars, faces)	DPs showed reduced face-selective responses in the bilateral FFA and smaller face-selective clusters in the r-FFA. A robust relationship between face selectivity and face identification ability was found in FG across all the samples.	DPs reported substantial life-long difficulties with face processing. DPs scored below (2 SD) the normal range on the FFT and the CFMT. DPs did not differ from HCs in IQ scores and low-level perceptual abilities.
Block fMRI	Gerlach et al., 2019 [60]	15 DPs, 34 HCs	Target detection task (faces, houses, tools, and words)	DPs showed reduced activation of core ventral face areas during perception of faces (more pronounced in the left hemisphere). No differences were found in activation to orthographic material and objects.	DPs reported substantial life-long difficulties with face processing. DPs scored below HCs in CFMT and a face identification task.
Block fMRI	Gilaie-Dotan et al., 2009 [61]	1 DP, 9 HCs	(1) Category-selectivity mapping (faces, houses, objects, and patterns); (2) motion-selectivity mapping (static, motion); (3) completion experiment	The secondary visual cortex (V2-V4) was strongly deactivated in DPs compared with HCs, whereas activity in the primary visual cortex and the VOTC was robust, with selectivity for faces and objects impaired mainly in the left hemisphere.	Face processing was extremely impaired in the DP case; no other kind of low- and high-level vision deficits were observed.
Block fMRI	Hasson et al., 2003 [50]	1 DP, 12 HCs	One-back task (faces, buildings, objects, geometric patterns); Exp. 2: Passive view (modified Rubin’s face–vase, lines drawing a face, and goblets)	DP’s face-related activation pattern in the VOTC was similar to that observed in HCs.	DP case was impaired on the FFT but was able to identify gender, age, and emotional state based on a person’s face. He had no difficulty in recognizing objects and exhibited normal performance in holistic and analytic visual, memory, and cognitive tests.
Block fMRI	Hadjikhai & de Gelder, 2002 [62]	1 DP, 2 HCs	Passive viewing (faces, objects, houses, scenes)	DP showed no face-selective activation in the FFA and OFA.	N/A.
Block fMRI	Jiahui et al., 2020 [63]	12 DPs, 16 HCs	One-back task	FFA and OFA responded strongly both in HCs and DPs when attention was directed to identity and expression, while pSTS and IFG responded the most when attention was directed to facial expression.	DPs reported problems with face recognition in daily life and scored below (1 SD) the normal range on at least 3 different diagnostic tests (CFMT, FFT, and an O/NFT). Expression recognition ability was preserved.
Block fMRI (dynamic causal modeling)	Lohse et al., 2016 [64]	15 DPs, 15 HCs	Passive viewing with repetitions (emotional faces and cars)	DPs exhibit reduced strength of feed-forward connections carrying face information from the early visual cortex to FFA and pSTS. These network alterations contribute to the diminished face selectivity in the posterior occipitotemporal areas affected in DP. This response profile was comparable in DPs and HCs.	DPs reported substantial life-long difficulties with face processing. DPs scored below (2 SD) the normal range on the FFT and the CFMT. DPs did not differ from HCs in IQ scores and low-level perceptual abilities.
Block fMRI	Minnebusch & Daum, 2009 [65]	4 DPs, 7 HCs	Passive viewing (famous, non-famous faces, caricatures and non-face objects)	HCs showed higher face-related activations in the r- compared to the l-FFA, with higher activation in the OFA and FFA for new faces vs. known faces. Contrary to HCs, DPs did not show bilateral face-related activations in the OFA and FFA.	DPs face recognition was heterogeneously impaired with normal basic visual processing and intact object recognition abilities.
Block fMRI	Németh et al., 2015 [66]	3 DPs, 20 HCs	One-back task on faces and nonsense objects and verbal report of the total number of one-back repetitions at the end of each run (total runs = 6)	FFA and OFA activity, as well as LOC activity, was significantly reduced in DPs. Analysis of the hemodynamic response function revealed a lower peak response, but also a significantly faster and stronger decay of the VOTC response in DPs.	DPs showed impaired face recognition and perception capacities on the CFMT, whereas object recognition and IQ were in the normal range.
Block fMRI (multi-item discrimability pattern)	Tian et al., 2020 [67]	64 DPs, 62 HCs	One-back task (faces, scenes, objects, and scrambled objects)	DPs’ r-FFA and OFA activation patterns for faces differed from the mean activation pattern of HCs.	DPs demonstrated face-specific impairments during 4 phases of a multiple-stage procedure, while low-level vision, multimodal person recognition, and general object recognition were intact. In the last stage, DPs scored below (1 SD) the normative range on a computer-based O/NFT and FFT.
Block fMRI (inter-subject functional correlation)	Rosenthal et al., 2017 [68]	10 DPs, 10 HCs	Passive viewing (emotional faces, neutral faces, familiar faces, unfamiliar faces, and non-faces)	The aTL served as the major network hub for face processing in HCs but not in DPs, which showed hyper-connectivity in lateral occipital and the inferior temporal cortices. The extent of this hyper-connectivity was correlated with DPs’ face recognition deficit.	DPs scored below (2 SD) the normal range at least on 2 out of 4 different diagnostic tests (FFT, CFMT, CFPT, discrimination of novel upright and inverted faces).
Block fMRI (multi-voxel pattern analysis)	Rivolta et al., 2014 [69]	7 DPs, 10 HCs	One-back task (faces, headless bodies, body parts, and objects)	Neural activity within the core and extended face regions in DPs showed reduced discriminability between faces and objects. Reduced differentiation between faces and objects in DP was also observed in the right parahippocampal cortex.	DPs scored below (2 SD) the normal range on at least 1 out of 3 diagnostic tests (FFT, CFMT and CFPT). DPs did not differ from HCs in IQ scores, low-level perceptual abilities, and object recognition.
Block fMRI (multi-voxel pattern analysis)	Zhang et al., 2015 [70]	7 DPs, 21 HCs	Passive viewing (intact face, face features, scrambled face, and non-face stimuli)	Right FFA’s responded preferably to faces in DPs and HCs, but no distinct neural response patterns were observed in DPs for the intact and the scrambled face configurations.	DPs scored below (2 SD) the normal range on a computer-based O/NFDT and FFT.
Block fMRI (multi-voxel pattern analysis)	Zhao et al., 2022 [71]	64 DPs, 62 HCs	Passive viewing with repetitions (faces, objects, scenes, or scrambled objects)	FFA in DPs showed attenuated repetition suppression for faces, suggesting an inefficient perceptual analysis for learned faces. At the mnemonic level, DPs showed decreased stability for repeated faces in MTG, suggesting an unstable mnemonic representation for learned faces, which was associated with impaired face recognition performance in DP.	DPs demonstrated face-specific impairments during 4 phases of a multiple-stage procedure, while low-level vision, multimodal person recognition, and general object recognition were intact. In the last stage, DPs scored below (1 SD) the normative range on a computer-based O/NFT and FFT.
Block fMRI (voxel-wise brain–behavior correlation analyses)	Liu et al., 2021 [72]	64 DPs, 61 HCs	Passive viewing (faces, objects, scenes, and scrambled objects)	DPs’ face memory performance was linked to bilateral FFA activity, while right pSTS activity was associated with face perception. Deficits in both tasks shared neural substrates in r-precuneus and r-orbitofrontal cortex.	DPs demonstrated face-specific impairments during 4 phases of a multiple-stage procedure, while low-level vision, multimodal person recognition, and general object recognition were intact. In the last stage, DPs scored below (1 SD) the normative range on a computer-based O/NFT and FFT.
Diffusor Tensor Imaging	Gomez et al., 2015 [73]	18 DPs, 18 HCs	None	DPs expressed an atypical tract structure-behavior relationship near face-selective regions.	DPs scored below (2 SD) the normal range on the CFMT.
Diffusor Tensor Imaging	Grossi et al., 2014 [74]	1 DP, 7 HCs	None	The right inferior longitudinal fasciculus was markedly reduced in DPs.	DP case scored borderline on the BFRT and was able to refer the faces’ gender but was not able to distinguish famous from unclear faces and could neither name nor identify celebrities from their photographs in the FFT. DP case showed mild visual agnosia for living and man-made objects. The general neuropsychological assessment did not reveal impairments.
Diffusor Tensor Imaging	Song et al., 2015 [75]	16 DPs, 16 HCs	None	ILF and IFOF white matter were comparable between DPs and HCs. DPs had lower fractional anisotropy in white matter local to the r-FFA.	DP case scored below (3 SD) the normal range on FFT.
Diffusor Tensor Imaging	Thomas et al., 2009 [52]	6 DPs, 12 HCs	None	DPs showed a marked reduction in the structural integrity of the inferior longitudinal fasciculus and the inferior fronto-occipito fasciculus bilaterally compared with HCs. In addition, DPs showed a reduction in fractional anisotropy in the bilateral FG, right anterior temporal stem, and left external capsule white matter.	DPs scored significantly lower than HCs in recognizing familiar faces and in making same/different discrimination judgments on unfamiliar faces. DPs were also impaired, although to a lesser extent and with greater variability, on tasks involving non-face stimuli. All DPs performed within the normal range on various low-level visual processing tasks.
Event-related fMRI	Avidan & Behrmann, 2009 [76]	6 DPs, 12 HCs	Familiarity identity judgment (famous vs. unknown X same vs. different)	The fMRI signal was greater in the HCs than in DPs, but normal face identity adaptation effects were observed in DPs. HCs, but not DPs, presented selective activation for familiar vs. unknown faces in the precuneus/posterior cingulate cortex and the anterior paracingulate cortex.	DPs scored significantly lower than HCs in recognizing familiar faces and making same/different discrimination judgments of unfamiliar faces. DPs were also impaired, although to a lesser extent and with greater variability, on tasks involving nonface stimuli. All DPs performed within the normal range on various low-level visual processing tasks.
Event-related fMRI	Haeger et al., 2021 [77]	13 DPs, 12 HCs	Modified Sternberg paradigm (low, medium and high load)	DPs failed to generate robust and maintained neural representations in the FFA during face encoding and maintenance.	The diagnosis was based on a complex pattern of features, representing both clinical complaints of long-term memory deficits and compensatory strategies, CFMT score, and family history. The CFMT revealed a significant difference in accuracy between DPs and HCs.
Event-related fMRI	Van den Stock et al., 2008 [78]	3 DPs, 4 HCs	Oddball detection task with pictures of fearful and happy expressions (body and faces), emotionally neutral body expressions, and houses	Neutral, but not emotional, faces triggered lower right FFA activation in the DPs compared with HCs. DPs showed stronger activation for bodies in the inferior occipital gyrus and for neutral faces in the extrastriate body area, indicating that body- and face-sensitive processes are less categorically segregated in DPs.	DPs scored below the normal range on the BFRT and/or the WFMT with a preserved visual recognition for non-face stimuli.
Functional connectivity fMRI (fractional amplitude of spontaneous low-frequency fluctuations and functional regional homogeneity)	Zhao et al., 2016 [79]	64 DPs, 62 HCs	Resting state	Different aspects of abnormal spontaneous neural activity within r-OFA underlie DP face-processing deficit.	DPs demonstrated face-specific impairments during 4 phases of a multiple-stage procedure, while low-level vision, multimodal person recognition and general object recognition were intact. In the last stage, DPs scored below (1 SD) the normative range on a computer-based O/NFT and FFT.
Functional connectivity fMRI	Avidan et al., 2014 [56]	7 DPs, 7 DPs	Visual stimulation/resting state	A typical connectivity pattern was observed in the core face network of the DPs, while diminished connectivity in the pSTS, r-aTC, and augmented connectivity of the r-amygdala were found in DPs compared with HCs.	DPs reported substantial life-long difficulties with face processing. DPs scored below (2 SD) the normal range at least at 2 out of 4 different experiments: the FFT, the CFMT, the CFPT, and a task measuring discrimination of novel upright and inverted faces.
Functional connectivity fMRI	Song et al., 2015 [75]	17 DPs, 17 healthy adults, 25 healthy children	Resting state	Core face network’s functional connectivity was disrupted in the DPs.	DPs scored below (3 SD) the normal range on FFT.
Functional connectivity fMRI	Zhao et al., 2022 [71]	64 DPs, 62 HCs	Resting state	Functional connectivity between the FFA and MTG was disrupted in DPs.	DPs demonstrated face-specific impairments during 4 phases of a multiple-stage procedure, while low-level vision, multimodal person recognition, and general object recognition were intact. In the last stage, DPs scored below (1 SD) the normative range on a computer-based O/NFT and FFT.
Functional connectivity fMRI (voxel-based global connectivity)	Zhao et al., 2018 [53]	64 DPs, 62 HCs	Resting state	Both the functional connectivity within the core face network and those between the core face network and extended face network were largely reduced in DPs. Importantly, the r-OFA and r-FFA served as the dysconnectivity hubs within the core face network. In addition, DPs’ r-FFA also showed reduced functional connectivity with the extended face network. This disrupted connectivity was related to DP’s face recognition deficit.	DPs demonstrated face-specific impairments during 4 phases of a multiple-stage procedure, while low-level vision, multimodal person recognition, and general object recognition were intact. In the last stage, DPs scored below (1 SD) the normative range on a computer-based O/NFT and FFT.
Structural MRI	Behrmann et al., 2007 [80]	6 DPs, 12 HCs	None	DPs evinced larger MTG and a significantly smaller aFG compared to HCs.	DPs scored significantly lower than HCs in recognizing familiar faces and making same/different discrimination judgments on unfamiliar faces. DPs were also impaired, although to a lesser extent and with greater variability, on tasks involving nonface stimuli. All DPs performed within the normal range on various low-level visual processing tasks.
Structural MRI	Bentin et al., 1999 [81]	1 DP, 15 HCs	None	DP case showed smaller right temporal lobes compared to HCs.	DP case showed normal visual perception, whereas performance with faces, although within normal range, was lower than HCs.
Structural MRI (voxel-based morphometry)	Garrido et al., 2009 [82]	17 DPs, 18 HCs	None	DPs had reduced grey matter volume in the r-aITG, STS/MTG bilaterally, and in the r-FG and r-aITG compared with HCs. Facial identity task performance correlated with l-STS/MTG and r-FG/ITG gray matter volumes.	DPs reported substantial life-long difficulties with face processing. DPs scored below (2 SD) the normal range on the FFT and the CFMT. DPs did not differ from HCs in IQ scores and low-level perceptual abilities.
Structural MRI	Gilaie-Dotan et al., 2009 [61]	1 DP, 9 HCs	None	No structural abnormalities in the DP case.	The DP case was strongly impaired in recognizing familiar and unfamiliar faces but fairly accurate at recognizing words, familiar places, and buildings.
Structural MRI	Van den Stock et al., 2012 [83]	1 DP, 20 HCs	None	Hypoplasia of the vermis cerebelli.	Low- and mid-level visual perception and object recognition abilities were intact in DP case, whereas face memory and recognition were impaired.
Structural MRI	Grossi et al., 2014 [74]	1 DP, 7 HCs	None	Mild cortical white matter atrophy (amygdala and hippocampal) in the bilateral medial temporal lobe.	The DP case scored borderline on the BFRT and was able to identify the faces’ gender but was not able to distinguish famous from unclear faces and could neither name nor identify celebrities from their photographs on the FFT. The DP case showed mild visual agnosia for living and man-made objects. The general neuropsychological assessment did not reveal impairments.
Structural MRI (voxel-based morphometry)	Dinkelacker et al., 2011 [58]	24 DPs, 25 HCs	None	DPs showed diminished gray matter density in the bilateral lingual gyrus, r-MTG, and l-DLPFC. In most of these areas, gray matter density correlated with memory success.	DPs were impaired in long-term memory for faces and, to a lesser extent, other complex visual stimuli such as buildings. However, their memory for faces with negative valence and their ability to categorize emotional expressions were preserved.
Structural MRI (voxel-based morphometry)	Haeger et al., 2021 [77]	13 DPs, 12 HCs	None	No structural abnormalities in the DP case.	DP diagnosis was based on a complex pattern of features, representing both clinical complaints of long-term memory deficits and compensatory strategies, CFMT score, and family history for some of the DP participants. The CFMT revealed a significant difference in accuracy between DPs and HCs.

**Table 3 brainsci-13-01399-t003:** EEG and MEG studies comparing brain activity in DPs and HCs. Abbreviations: FFT = Famous Face Test; CFMT = Cambridge Face Memory Test; CFPT = Cambridge Face Perception Test; BFRT = Benton Facial Recognition Test; WFMT = Warrington Face Memory Test; RMT = Recognition Memory Test; IQ = intelligence quotient.

Protocol/ Outcomes	Reference	Participants	Task Administered during the Protocol	Main Findings	Diagnosis and Behavioral Data
ERP (N170)	Gilaie-Dotan et al., 2009 [61]	1 DP, 9 HCs	Passive viewing (faces, watches, and flowers)	No face-selective N170 emerged in the DP subject.	The DP case was strongly impaired at recognizing familiar and unfamiliar faces but fairly accurate at recognizing words, familiar places, and buildings.
ERP (N170)	Bentin et al., 1999 [81]	1 DP, 12 HCs	The matching task for covert recognition (names/faces of politicians vs. movie stars)	No face-selective N170 emerged in the DP subject.	The DP case showed normal visual perception, whereas performance with faces, although within the normal range, was lower than HCs.
ERP (N170)	Bentin et al., 2007 [84]	1 DP, 12 HCs	One-back task (faces, places, and objects)	No face-selective responses emerged in the DP subject. As compared to HCs, the FFA of DPs responded more to objects than faces. No face-specific N170 emerged in the DPs.	The DP case scored lower than HCs in the FFT, CFMT, WFMT (but not words), BFRT, and Boston Naming Test (object recognition). The DP case performed well on the one-back task.
ERP (N170)	Burns et al., 2014 [85]	8 DPs, 11 HCs	“Remember/know” paradigm for recollection and familiarity	Delayed right posterior area’s N170 responses, unexpected frontal responses, and no posterior responses associated with familiarity for faces emerged in DPs.	DPs scored below (2 SD) the normal range on the CFMT and FFT and exhibited significantly worse performances on the “Remember/know” task than HCs.
ERP (N170)	Collins et al., 2017 [86]	7 DPs, 10 HCs	Face/word stimuli (participants were asked to respond if two sequentially presented stimuli were the same or not)	As compared to HCs, DPs did not exhibit the typical N170 hemispheric preference (i.e., left for words and right for faces).	DPs scored below (1.5 SD) the normal range on the CFMT and FFT. DPs performed worse than HCs in detecting the same faces but not the same words.
ERP (N170)	DeGutis et al., 2007 [57]	1 DP, 24 HCs	One-back task (faces, scenes); face classification training	DP’s N170 responses for pictures of faces and watches were comparable, while, during active training, a conspicuous selectivity emerged and resembled that observed in HCs.	The DP case was severely impaired in face recognition and classification, whereas she was within the normal range on the one-back face task and object and word recognition. Following training, the DP case’s performance on face classification task improved.
ERP (N170)	Kress & Daum, 2003 [87]	2 DPs	Classification of known/unknown faces, hand, and houses: hands direction task; face recognition test	Same-amplitude N170 for faces and houses in DPs.	One DP case scored below (2 SD) the normal range at the RMT, while the other one scored within the normal range. Both DPs exhibited poor performances on face recognition, while basic perceptual abilities and facial identity and affect discriminations were intact.
ERP (N170)	Lueschow et al., 2015 [88]	13 DPs, 16 HCs	Motion discrimination task; famous face/building recognition	Face-selective N170 was indistinguishable between DPs and HCs.	DPs scored lower than HCs on the BFRT, the WRMT, the CFMT, and the FFT, while their performances for house recognition were indistinguishable from HCs.
ERP (N170)	Minnebusch et al., 2007 [89]	4 DPs, 55 HCs	Viewing of stimuli from 16 categories, including famous/non-famous faces and various objects	Three out of 4 DPs showed reliable N170 amplitude differences between faces and nonface stimuli. One DP individual showed significantly reduced amplitude differences between faces and nonface objects.	DPs face recognition was heterogeneously impaired with normal basic visual processing and intact with object recognition abilities.
ERP (N170)	Németh et al., 2014 [90]	3 DPs, 20 HCs	Category (face–non face) discrimination task	Reduced DPs’ N170 sensitivity—potentially due to larger noise-elicited N170, rather than smaller face-elicited N170.	DPs showed impaired face recognition on CFMT, whereas object recognition and IQ were in the normal range.
ERP (N170)	Towler et al., 2012 [91]	16 DPs, 16 HCs	One-back task (faces and houses)	No face inversion effect (i.e., enhanced N170 for inverted stimuli) emerged in DPs.	DPs exhibited impaired performances on the CFMT, the FFT, and the O/NFT compared with HCs. There was also evidence for (upright) face perception deficits in the CFPT upright and inverted, while low-level visual perceptual abilities were normal
ERP (N170, P1)	Righart & de Gelder, 2007 [92]	4 DPs, 12 HCs	Orientation–decision task (upright versus inverted)	No configural encoding in 3 out of 4 DPs for faces at the P1component, and for both faces and bodies at the N170 component.	All DPs scored worse than HCs on the BFRT, while 3 out of 4 DPs scored below average at the WFMT.
ERP (N250)	[93]	12 DPs, 12 HCs	Identity/expression-matching task	Attenuated N250 correlating positively with CFMT emerged in DPs.	DPs scored below (2 SD) the normal range at least at 2 out of 3 different diagnostic tests (FFT, CFMT, CFPT) while performing within the normal range on the RMET. In addition, DPs performed worse than controls on the Identity-matching task, while no differences emerged from the expression-matching task.
ERP (N250)	[94]	14 DPs, 14 HCs	Face-matching task	Attenuated N250 emerged in DPs compared to HCs.	DPs scored below (2 SD) the normal range on the FFT, the O/NFT, and the CFMT. The majority of DPs performed within the normal range on the CFPT but worse than HCs on the face-matching task
ERP (N250, P600)	Eimer et al., 2012 [95]	12 DPs, 16 HCs	The task for familiarity judgment of famous and non-famous faces	Occipito-temporal N250 responses emerged in 6 out of 12 DPs. Non-recognized famous faces did not trigger a P600f component in 11 out of 12 DPs.	DPs scored below HCs on the CFMT, CFPT (with only one exception), O/NFT, and in famous and non-famous faces familiarity judgment.
ERPs	Parketny et al., 2015 [96]	10 DPs, 10 HCs	Target identification task	Reliable but delayed N250 and P600f emerged in DPs.	DPS scored below (2 SD) the normal range on the FFT, CFMT, CFPT, and O/NFT. Scores for the target identification task were poorer in DPs compared to HCs.
EEG summed amplitude values	Fisher et al., 2020 [97]	10 DPs, 10 HCs (exp. 1), 12 DPs, 12 HCs (exp. 2)	Oddball discrimination task with upright/inverted faces (exp.1) and faces/cars (exp.2)	Attenuated neural responses (SNS values) to oddball changes emerged in DPs compared to HCs (for both faces and cars).	DPs scored below (2 SD) the normal range on the FFT, the CFMT, and the O/NFT. Performance on the CFPT was more heterogeneous. DPs performed within the normal range on the old/new car memory test.
ERPs (source reconstructions)	Olivares et al., 2021 [98]	1 DP, 14 HCs	The face-feature-matching task with unknown faces, including external (E-) and internal (I-) facial features presented sequentially, then followed by complete matching (correct combination of I- and E-features) or mismatching (different I- and E-features) unfamiliar faces (i.e., targets)	More positive waveforms in matching faces than mismatching ones ~300–500 ms in the E-I condition (i.e., a ‘mismatch effect’) emerged in HCs only. Typical P1-N170 for I-features, but no P2, was found in the DP case; a mismatch effect emerged in the DP patient for the I-E sequence (of shorter duration than that of HCs). Differential topological activations emerged between HCs and DPs: posterior and markedly left-sided (around the OFA) in the DP case, posterior and right-sided in HCs. No P3 responses in response to features were found in the DP case.	The DP case exhibited normal QI and face-specific and associative impairment (i.e., no object processing and/or apperceptive impairments). Performances on the face-feature-matching task revealed no significant differences from HCs but a tendency towards erroneous positive “match” responses in both sequences (significant in the E-I condition).
Gamma-band activity (MEG source analysis)	Dobel et al., 2011 [99]	7 DPs, 7 HCs	Recognition of faces differing in familiarity (famous vs. unknown) and orientation (upright vs. inverted)	No increase in gamma-band activity in the left lateral occipitotemporal gyrus and the left inferior temporal gyrus in response to faces emerged in DPs compared to HCs.	DPs showed normal general visual functions, while performances on the FFT differed significantly from those of HCs. DPs identified significantly less famous faces than HCs in the experimental task for recognition.
M170	Lueschow et al., 2015 [88]	13 DPs, 16 HCs	Motion detection task, including pictures of faces and houses	Comparable M170 responses to objects emerged in DPs and HCs, while faces elicited prolonged latency and decreased amplitude M170s in DPs. Correlation between face recognition performance and the size of the M170 emerged in HCs but not DPs.	DPs and HCs differed in IQ assessment; however, most of the results from neuropsychological tests were indistinguishable between DPs and HCs. DPs scored significantly lower than HCs on the BFRT, the WRMT, the FFT, and the CFMT, while their performances for houses recognition were analogous to HCs.
M170	Rivolta et al., 2012 [100]	6 DPs, 11 HCs	Target detection task, including pairs of famous/non-famous faces and places	Greater MEG activity for faces than places (category effect) emerged in the r-LOC and r-FG (i.e., M170 generators) in both HCs and DPs. r-LOC-M170 correlated with holistic/configural face processing, whereas the r-FG-M170 correlated with featural processing.	DPs scored below (2 SD) the normal range on the CFMT and FFT. DPs did not differ from HCs in IQ scores, low-level perceptual abilities, and object recognition, and none of them scored within the autistic range.
M170 (neural sources estimation via L2-MNE approach)	Dobel et al., 2008 [101]	7 DPs, 7 HCs	Recognition of faces differing in familiarity (famous vs. unknown) and orientation (upright vs. inverted)	A bilateral decrease in brain activity in the initial phase of the M170 merged in DPs compared to HCs, followed by an even larger reduction over occipitotemporal areas in the left hemisphere (i.e., lateralization effect), irrespectively of familiarity or orientation of the stimuli.	DPs showed normal general visual functions, while their performances on the FFT differed significantly from those of HCs. DPs identified significantly less famous faces than HCs in the experimental task.
M170 + N170	Harris et al., 2005 [102]	5 DPs, 8 HCs	Viewing of pictures, including faces, houses, and miscellaneous objects	M170 was not face-selective in 3 of 5 DPs. ERPs in the remaining DPs showed N170s within the same normal range.	Only one DP exhibited impaired performances on visual function tests. All DPs scored 2 SD below the mean on the FFT and between in the O/NFT.

## Data Availability

No new data were created or analyzed in this study. Data sharing is not applicable to this article.

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
