# Peer review of "The Neural Correlates of Developmental Prosopagnosia: Twenty-Five Years on"

_brainsci, 2023, doi:10.3390/brainsci13101399_

Round 1

Reviewer 1 Report

In the present manuscript, the Authors investigate the neural correlates of developmental prosopagnosia, by reviewing structural and functional studies.  I have to say that the paper addresses a very relevant and needed issue in the face literature. The paper is well-written and the review is very comprehensive. I just have some minor details:

- I would try to reduce the number of abbreviations; I liked that the authors included a glossary of those, but where possible I would substitute them with the full words (i.e., FC, CP, etc.)

- Should figure 2 reference to Hxby et al., 2000?

- It seems that the Authors are referring a lot to the holistic/configural hypotheses in prosopagnosia. However, this is not the only one (i.e., the expertise hypothesis) so the Authors should town down a little this part. Also what the link between reduced grey/white matter or reduced connectivity between two brain areas and the holistic deficit is not clear (and mostly speculative at this point)

- Sections 5 needs some references (e.g., Barton, The problem of being bad at faces - and also line 358)

- I am not sure that the two hypotheses mentioned in the conclusions are exclusive of each other - again, I think that there is too much focus on the holistic interpretation

none

Author Response

In the present manuscript, the Authors investigate the neural correlates of developmental prosopagnosia, by reviewing structural and functional studies. I have to say that the paper addresses a very relevant and needed issue in the face literature. The paper is well-written and the review is very comprehensive. I just have some minor details:

COMMENT 1: I would try to reduce the number of abbreviations; I liked that the authors included a glossary of those, but where possible I would substitute them with the full words (i.e., FC, CP, etc.)

RESPONSE 1: We acknowledge this issue; thus, we have significantly minimized the use of abbreviations throughout the text, enhancing readability and improving the overall content flow. We have retained some of the previous abbreviations in the tables, explaining them in the caption.

COMMENT 2: Should figure 2 reference to Haxby et al., 2000?

RESPONSE 2: Yes, figure 2 is a revision of the figure of Haxby et al., which we have now cited in the caption.

COMMENT 3: The Authors seem to refer a lot to the holistic/configural hypotheses in prosopagnosia. However, this is not the only one (i.e., the expertise hypothesis) so the Authors should town down a little this part. Also the link between reduced grey/white matter or reduced connectivity between two brain areas and the holistic deficit is not clear (and mostly speculative at this point)

RESPONSE 3: Based on the Reviewer’s concern, the Introduction has been updated with an overview of the face processing hypotheses (i.e., holistic, featural/configural, and expertise hypotheses; pages 1-2, lines 35-48), and the expertise hypothesis is now mentioned in the conclusions.

The prevalent reference to the holistic/configural interpretation in the context of this paper stems from previous literature and the reviewed studies in which the holistic/configural interpretation of face processing is dominant. Furthermore, the expertise hypothesis has been challenged in the context of prosopagnosia (e.g., Rezlescu et al., 2014 https://www.pnas.org/doi/abs/10.1073/pnas.1317125111; Duchaine et al., 2004, https://www.sciencedirect.com/science/article/pii/S0896627304004921). As highlighted by Davies-Thompson et al. (2014) (https://pubmed.ncbi.nlm.nih.gov/24389150/) the assessment of this hypothesis should include the evaluation of prosopagnosics’ proficiency with various object types before and after the onset of their symptoms, which is challenging per se and not possible in DP. Furthermore, none of the reviewed studies include the assessment of patients’ performances with Greebles, which has been previously adopted to test the expertise hypothesis.

Given the Reviewer’s concern on the link between reduced connectivity and holistic deficit in DP, this interpretation has been revised considering a more cautious perspective (page 18-19).

COMMENT 4: Sections 5 needs some references (e.g., Barton, The problem of being bad at faces – and also line 358)

RESPONSE 4: Thanks for this suggestion. In the revised version of the manuscript, we have added this reference and others.

COMMENT 5: I am not sure that the two hypotheses mentioned in the conclusions are exclusive of each other - again, I think that there is too much focus on the holistic interpretation

RESPONSE 5: In light of the Reviewer’s concern, the section discussing the available hypothesis on the cognitive bases of prosopagnosia has been extensively updated (pages 18-19 lines 442-451).

Reviewer 2 Report

This review provides a welcome overview of structural and functional neuroimaging studies of congenital prosopagnosia (CP) over the last 25 year, and will make a fine contribution to the special issue. I enjoyed reading the paper. There are, however, some major issues I think the authors should address before publication, and a number of smaller comments.

Major issues:

1.       It is not clear exactly what kind of review the authors have conducted. Although they mention on p.3 that it is a scoping review, they cite no guidelines for conducting such a review and importantly do not report any evaluation of the quality of the included studies.

I think a quality-assessment could be helpful in determining the strength of the evidence for the different conclusions drawn by the authors. At least this should be considered, and if the authors choose not to conduct a quality assessment, the rationale for this should be explained. Relatedly, it could be made clearer earlier in the paper what the overall aim of the review is. Lines 82-84 (p. 2) presents an aim to shed light on the convergent and contrasting findings in the CP neuroimaging literature, but it would be helpful to elaborate on this aim. Is the aim to synthesize the evidence on functional and anatomical differences between CP’s and controls? Also, it would be good to present this aim earlier in the introduction.

2.       There is a mix of cognitive and anatomical arguments and interpretations, that I found confusing. The authors readily infer from cognition to anatomical locations and from anatomical locations to cognitive functions, and in many places without acknowledging that most of the connections between structure (anatomical location) and function (e.g. face detection or perception) is hypothetical not factual. Few if any studies address specific levels of processing in prosopagnosia (or neurotypical face processing) to underlying anatomical structures, and there is a lot we do not yet know about this relationship. This should be acknowledge in the paper, and the conclusions tempered accordingly, so that causal conclusions about cognition are not suggested based on anatomical findings and vice versa.  (one example: p.11., line 172-177).This does not detract from the value of the paper, which is a welcome contribution to the literature, but will make it more helpful in keeping the anatomical and cognitive hypotheses and conclusions apart. I suggest reading the whole manuscript with this in mind and revise the paper to 1) keep structural and cognitive inferences clearly separate/distinct, and 2) (This is more a suggestion) focus on the anatomical/neuroimaging differences between participants with CP and healthy controls, which is the aim of the paper. Or summarize the anatomical findings first, and then link anatomy to function in a separate section.

In the conclusion, this mix of cognitive and anatomical levels of argument/inference is particularly problematic, and I suggest rephrasing the conclusion to better reflect the findings of the imaging studies, rather than focusing on cognitive inferences that are relatively loosely based on the reported findings.

3.       Lateralization has been a key point of dispute in the literature on face recognition and prosopagnosia. I was surprised to see that the review has relatively little focus on this, and in many places anatomical regions are mentioned without indication of laterality. This could be improved, perhaps by including a section in the discussion about laterality if (dys)function and structural abnormalities in CP.

Other issues:

4.       The special issue title is ‘Insights into Developmental Prosopagnosia’, and it seems appropriate that the authors use this term rather than Congenital Prosopagnosia (CP). If they choose to stick with CP, an argument should be presented for this, and the debate about taxonomy briefly mentioned. Perhaps the authors have an opinion here.

5.       I am not an imaging expert, but I know that in some commonly used MRI sequences, there is signal dropout in the anterior temporal lobes and also the medial temporal lobe. Does this have any bearing on the conclusions that can be drawn from the reviewed studies, and is it worth mentioning? Some of the included studies have likely corrected for this in various ways while other probably have not.

6.       Regarding Tables 2 and 3: It would be very helpful if the author names and publication year of the studies (e.g., with an APA style reference (e.g. Avidan et al., 2005) were included in addition to the existing “Reference” column (e.g. [47].

7.       Regarding Tables 2 and 3: It could be made a bit clearer what the column “Task” represents.

8.       Regarding Table 2, row 2: under “Participants”, it should probably say ‘7 CP, 7 HC’?

9.       Please make it clearer in the text when you are referring the results of a single study and when you are summarizing results from multiple studies.

10.   The possible heterogeneity of CP is mentioned in various places in the manuscript, e.g., p. 12, line 232; p 15, line 340. This heterogeneity could reflect that the condition is not homogenous, or be a problem of definition and diagnosis (which plagues the field). As it is, we do not know what the main cause is. This could be tackled more specifically in the discussion.

11.   Small thing, but what is a ‘disconnectivity hub’ (p. 10, line 125)? I get that connections have hubs, but disconnectivity seems to imply there isn’t one?

12.   In lines 360-367 (p. 16), the authors conclude that there is strong support for the hypothesis that face processing in CP happens in a ‘non-face’ manner. It might be worth bringing in Geskin & Behrmann’s 2018 review of object processing in CP here (DOI: 10.1080/02643294.2017.1392295).

13.   In general, the Conclusion could better reflect the ambiguity of the evidence nicely presented in the Discussion.

14.   It might work well if the paper ends up with presenting future directions for resolving some of the issues/inconsistencies presented in the review.

15.   P 12, lines 219-222: it is unclear what this sentence is meant to convey – please rephrase.

16.   P. 14, lines 273-283; This paragraph is messy and difficult to read, and should be rewritten.

17.   P. 16, line 356; ‘at least some of the’ instead of ‘the emerged’.

1. There are very many abbreviations in the paper, many of which are not common, and which precludes fluent reading of the paper. Please consider if some of these could just be spelled out in the manuscript. For example abbreviations like CFN and EFN are not commonly known outside the field, ans will be challenging to understand for many readers, so please spell out core face network and extended face network throughout. Some abbreviations are not explained in the text or Abbreviation glossary (e.g., FC).

2. The manuscript shuold be thoroughly proof read, with special attention to verb tense (use the present tense for the work of the review; e.g., p. 2, line 95 'summarize' instead of 'summarized'; like 97; sample of CPs analyzed is' not 'was' - please check this throughout the ms.

3. The word 'revise' is used instead of 'review' - please change this (search the whole document). Examples are p. 2, line 83; p. 10, line 115 - but there are more.

4. The word consistenstly is used inappropriately, and most sentences containing this word should be revised (search the whole document). E.g., p. 10, like 115 "Most ...studies consistently reported' - if not all studies reported something, I do not think it qualifies as consistent?

5. CP particpants are reported as 'CP patients' in many places, particularly towards the end of the manuscript. They are not patients, and should not be referred to as such. Use participants or a similarly neutral wording. Possibly the Journal has guidelines for language that could be followed, or chec the APA-recommendations on unbiased language.

Author Response

This review provides a welcome overview of structural and functional neuroimaging studies of congenital prosopagnosia (CP) over the last 25 year, and will make a fine contribution to the special issue. I enjoyed reading the paper. There are, however, some major issues I think the authors should address before publication, and a number of smaller comments.

Major issues:

COMMENT 1: It is not clear exactly what kind of review the authors have conducted. Although they mention on p.3 that it is a scoping review, they cite no guidelines for conducting such a review and importantly do not report any evaluation of the quality of the included studies. I think a quality-assessment could be helpful in determining the strength of the evidence for the different conclusions drawn by the authors. At least this should be considered, and if the authors choose not to conduct a quality assessment, the rationale for this should be explained.

Relatedly, it could be made clearer earlier in the paper what the overall aim of the review is. Lines 82-84 (p. 2) presents an aim to shed light on the convergent and contrasting findings in the CP neuroimaging literature, but it would be helpful to elaborate on this aim. Is the aim to synthesize the evidence on functional and anatomical differences between CP’s and controls? Also, it would be good to present this aim earlier in the introduction.

RESPONSE 1: We thank the Reviewer for pointing out these weaknesses. To provide more clarity on the nature of our review, we have now explicitly mentioned that we conducted a scoping review following the guidelines outlined in Peters et al. (2015) and Munn et al. (2018). Notably, these guidelines do not specifically recommend the inclusion of a quality assessment, such as a risk of bias evaluation, for the included studies. Furthermore, it's important to consider the complexity of our review, which encompasses various neuroimaging techniques; conducting a uniform quality assessment in this context could be challenging and may not yield meaningful insights given the diversity of methodologies and study designs. Instead, we have taken a strategic approach by describing every single study in the proper tables and placing a stronger emphasis in the discussion sections (i.e., 3, 4) on data derived from more rigorous studies or findings that have been consistently observed across multiple studies. We believe that this approach allows us to highlight the most robust and reliable evidence while also recognizing the limitations of less rigorous studies without imposing a fictitious classification.

Further, we have revised the aim statement in the Introduction section to clarify the review goals (page 3, lines 96-99).

COMMENT 2: There is a mix of cognitive and anatomical arguments and interpretations, that I found confusing. The authors readily infer from cognition to anatomical locations and from anatomical locations to cognitive functions, and in many places without acknowledging that most of the connections between structure (anatomical location) and function (e.g. face detection or perception) is hypothetical not factual. Few if any studies address specific levels of processing in prosopagnosia (or neurotypical face processing) to underlying anatomical structures, and there is a lot we do not yet know about this relationship. This should be acknowledged in the paper, and the conclusions tempered accordingly, so that causal conclusions about cognition are not suggested based on anatomical findings and vice versa.  (one example: p.11., line 172-177). This does not detract from the value of the paper, which is a welcome contribution to the literature, but will make it more helpful in keeping the anatomical and cognitive hypotheses and conclusions apart. I suggest reading the whole manuscript with this in mind and revise the paper to 1) keep structural and cognitive inferences clearly separate/distinct, and 2) (This is more a suggestion) focus on the anatomical/neuroimaging differences between participants with CP and healthy controls, which is the aim of the paper. Or summarize the anatomical findings first, and then link anatomy to function in a separate section.

In the conclusion, this mix of cognitive and anatomical levels of argument/inference is particularly problematic, and I suggest rephrasing the conclusion to better reflect the findings of the imaging studies, rather than focusing on cognitive inferences that are relatively loosely based on the reported findings.

RESPONSE 2: We thanks to the Reviewer for this valuable suggestion. We have realized that the “conclusions” section was focused on the cognitive processes underlying prosopagnosia, rather than on summarizing the neuroimaging studies considered. In the revised version of the manuscript, we have extensively restructured and rewritten the conclusions in line with the reviewer's suggestion (pages 18-19).

COMMENT 3: Lateralization has been a key point of dispute in the literature on face recognition and prosopagnosia. I was surprised to see that the review has relatively little focus on this, and in many places anatomical regions are mentioned without indication of laterality. This could be improved, perhaps by including a section in the discussion about laterality if (dys)function and structural abnormalities in CP.

RESPONSE 3: We agree with the Reviewer's concern. Indeed, we only briefly mentioned the right hemisphere dominance for faces in the introductions, but we did not elaborate on it in the discussions. In the revised version of the manuscript, we have now added a paragraph that delves deeper into this question, providing a more comprehensive discussion of laterality in the context of (dys)function and structural abnormalities in DP (page 18, lines 428-447)).

Other issues:

COMMENT 4: The special issue title is ‘Insights into Developmental Prosopagnosia’, and it seems appropriate that the authors use this term rather than Congenital Prosopagnosia (CP). If they choose to stick with CP, an argument should be presented for this, and the debate about taxonomy briefly mentioned. Perhaps the authors have an opinion here.

RESPONSE 4: Thanks to the reviewer for this valuable suggestion. We usually use “congenital” or “developmental” as interchangeable terms; thus, in the revised version of the manuscript we opted for “developmental prosopagnosia” (in line with the special issue title).

COMMENT 5:  I am not an imaging expert, but I know that in some commonly used MRI sequences, there is signal dropout in the anterior temporal lobes and also the medial temporal lobe. Does this have any bearing on the conclusions that can be drawn from the reviewed studies, and is it worth mentioning? Some of the included studies have likely corrected for this in various ways while other probably have not.

RESPONSE 5: We acknowledge the Reviewer's concern on the potential impact of signal dropout and distortions on temporal lobe imaging in commonly used MRI sequences, as reported in previous studies (e.g., doi:10.1006/nimg.1997.0289; 4, 101–109, doi:10.1093/scan/nsn044). There are at least two strategies researchers can use to limit this “danger”: (i) adopting a block design that, as opposed to an event-related design, improves the signal-to-noise ratio and/or (ii) implementing a multivariate analysis (rather than a mass-unibariate analysis). Much of the evidence reported in this review adopts at least one of the two strategies (some adopt both), which makes us confident of the overall good reliability of our conclusions/discussions (see page 18, lines 391-397). The role of anteror lobe in face perception is also supported by neurophysiological data (e.g., doi: 10.1227/NEU.0000000000000789) and clinical data, where some forms of acquired prosopagnosia have been linked to anterior temporal lobe injury or neurodegenerative processes (e.g., doi: 10.2147/EB.S92838; doi: 10.1097/WAD.0000000000000511). Therefore, while concerns about detecting anterior temporal lobe signals in fMRI studies could be considered, the extensive body of literature strongly supports the involvement of the anterior temporal lobe (a part of the extended face network) in both acquired and developmental forms of prosopagnosia.

COMMENT 6: Regarding Tables 2 and 3: It would be very helpful if the author names and publication year of the studies (e.g., with an APA style reference (e.g. Avidan et al., 2005) were included in addition to the existing “Reference” column (e.g. [47].

RESPONSE 6: We revised the tables according to the reviewer's suggestion

COMMENT 7: Regarding Tables 2 and 3: It could be made a bit clearer what the column “Task” represents.

RESPONSE 7: We have replaced the previous column designation with “Task administered during the protocol”. We hope that is clearer.

COMMENT 8: Regarding Table 2, row 2: under “Participants”, it should probably say ‘7 CP, 7 HC’?

RESPONSE 8: We have revised this typo, thanks for pointing out.

COMMENT 9: Please make it clearer in the text when you are referring the results of a single study and when you are summarizing results from multiple studies.

RESPONSE 9: We have deliberately chosen to emphasize findings rather than studies/authors throughout the main text to improve the readability and flow of the manuscript. Often, before discussing a particular result or evidence, the sentence is introduced with terms (such as “Most of the reviewed evidence…”) or with specific references (“For instance, Haeger et al. and Gilaie-Dotan et al. [61,77]”) that provide hints about whether it is based on multiple studies or single studies.

In addition, the description of each finding is followed by the corresponding reference(s) in brackets. When a singular reference is cited, it means that the result is based on a single study. Conversely, when multiple references are listed at the end of a statement, it indicates that the result has been consistently observed across various studies.

Obviously, in sections 3 and 4, there was a stronger emphasis on data derived from rigorous single studies and multiple studies, while results from single cases or less rigorous studies, were only briefly mentioned.

We hope that the reviewer agrees with this choice, which helps make the manuscript's reading smoother.

COMMENT 10: The possible heterogeneity of CP is mentioned in various places in the manuscript, e.g., p. 12, line 232; p. 15, line 340. This heterogeneity could reflect that the condition is not homogenous, or be a problem of definition and diagnosis (which plagues the field). As it is, we do not know what the main cause is. This could be tackled more specifically in the discussion.

RESPONSE 10: According to the Reviewer’s comment, a section highlighting the controversy of prosopagnosia etiology has been included in section 6 (page 18 lines 300-408).

COMMENT 11: Small thing, but what is a ‘disconnectivity hub’ (p. 10, line 125)? I get that connections have hubs, but disconnectivity seems to imply there isn’t one?

RESPONSE 11:  For "disconnectivity hub" we refer to the specific disrupted connectivity between r-OFA and r-FFA frequently observed in DPs. We have revised this sentence to improve its clarity (page 11 line 148).

COMMENT 12: In lines 360-367 (p. 16), the authors conclude that there is strong support for the hypothesis that face processing in CP happens in a ‘non-face’ manner. It might be worth bringing in Geskin & Behrmann’s 2018 review of object processing in CP here (DOI: 10.1080/02643294.2017.1392295).

RESPONSE 12: Thanks for this suggestion. In the revised version of the manuscript, we have added this reference.

COMMENT 13: In general, the Conclusion could better reflect the ambiguity of the evidence nicely presented in the Discussion.

RESPONSE 13: Section 6 has been improved according to the Reviewer’s comment (pages 18-19)

COMMENT 14: It might work well if the paper ends up with presenting future directions for resolving some of the issues/inconsistencies presented in the review.

RESPONSE 14: Based on the Reviewer’s comment, a paragraph highlighting future directions has been added to the Conclusions (page 19 lines 465-480).

COMMENT 15: P 12, lines 219-222: it is unclear what this sentence is meant to convey – please rephrase.

RESPONSE 15: We have rephrased this sentence (page 13 lines 227-234)

COMMENT 16: P. 14, lines 273-283; This paragraph is messy and difficult to read, and should be rewritten.

RESPONSE 16: In light of the Reviewer’s concern, this section has been rewritten (page 16 lines 304-314).

COMMENT 17: P. 16, line 356; at least some of the’ instead of ‘the emerged’.

RESPONSE 17: We have replaced “the emerged” with “at least some of the” as suggested by the reviewer.

Comments on the Quality of English Language

COMMENT 18: There are very many abbreviations in the paper, many of which are not common, and which precludes fluent reading of the paper. Please consider if some of these could just be spelled out in the manuscript. For example abbreviations like CFN and EFN are not commonly known outside the field, and will be challenging to understand for many readers, so please spell out core face network and extended face network throughout. Some abbreviations are not explained in the text or Abbreviation glossary (e.g., FC).

RESPONSE 18: We acknowledge the Reviewer for this suggestion. We strongly reduced the number of abbreviations throughout the text, enhancing readability and improving the overall flow of the content. We have retained some of the previous abbreviations only in the tables, explaining them in the caption.

COMMENT 19: The manuscript should be thoroughly proof read, with special attention to verb tense (use the present tense for the work of the review; e.g., p. 2, line 95 'summarize' instead of 'summarized'; like 97; sample of CPs analyzed is' not 'was' - please check this throughout the ms.

COMMENT 20: The word 'revise' is used instead of 'review' - please change this (search the whole document). Examples are p. 2, line 83; p. 10, line 115 - but there are more.

COMMENT 21: The word consistenstly is used inappropriately, and most sentences containing this word should be revised (search the whole document). E.g., p. 10, like 115 "Most ...studies consistently reported' - if not all studies reported something, I do not think it qualifies as consistent?

COMMENT 22: CP particpants are reported as 'CP patients' in many places, particularly towards the end of the manuscript. They are not patients, and should not be referred to as such. Use participants or a similarly neutral wording. Possibly the Journal has guidelines for language that could be followed, or chec the APA-recommendations on unbiased language.

RESPONSE 19,20,21,22: We have revised the manuscript based on these suggestions. We would like to thank the reviewer for helping us improve the quality of the manuscript.

Round 2

Reviewer 2 Report

The authors have addressed my comments in a very comprehensive way. Look forward to seeing the paper published.